# Cross Talk between Hydrogen Peroxide and Nitric Oxide in the Unicellular Green Algae Cell Cycle: How Does It Work?

**DOI:** 10.3390/cells11152425

**Published:** 2022-08-05

**Authors:** Wojciech Pokora, Szymon Tułodziecki, Agnieszka Dettlaff-Pokora, Anna Aksmann

**Affiliations:** 1Department of Plant Physiology and Biotechnology, Faculty of Biology, University of Gdańsk Wita, Stwosza 59, 83-308 Gdańsk, Poland; 2Department of Biochemistry, Medical University of Gdańsk, Dębinki 1, 80-211 Gdańsk, Poland

**Keywords:** hydrogen peroxide, nitric oxide, algae, cell cycle, *Chlamydomonas*

## Abstract

The regulatory role of some reactive oxygen species (ROS) and reactive nitrogen species (RNS), such as hydrogen peroxide or nitric oxide, has been demonstrated in some higher plants and algae. Their involvement in regulation of the organism, tissue and single cell development can also be seen in many animals. In green cells, the redox potential is an important photosynthesis regulatory factor that may lead to an increase or decrease in growth rate. ROS and RNS are important signals involved in the regulation of photoautotrophic growth that, in turn, allow the cell to attain the commitment competence. Both hydrogen peroxide and nitric oxide are directly involved in algal cell development as the signals that regulate expression of proteins required for completing the cell cycle, such as cyclins and cyclin-dependent kinases, or histone proteins and E2F complex proteins. Such regulation seems to relate to the direct interaction of these signaling molecules with the redox-sensitive transcription factors, but also with regulation of signaling pathways including MAPK, G-protein and calmodulin-dependent pathways. In this paper, we aim to elucidate the involvement of hydrogen peroxide and nitric oxide in algal cell cycle regulation, considering the role of these molecules in higher plants. We also evaluate the commercial applicability of this knowledge. The creation of a simple tool, such as a precisely established modification of hydrogen peroxide and/or nitric oxide at the cellular level, leading to changes in the ROS-RNS cross-talk network, can be used for the optimization of the efficiency of algal cell growth and may be especially important in the context of increasing the role of algal biomass in science and industry. It could be a part of an important scientific challenge that biotechnology is currently focused on.

## 1. Introduction

The role of reactive oxygen species (ROS) and reactive nitrogen species (RNS) in plant cell response to biotic and abiotic stresses is well documented [1,2,3]. Most of this discussion focusses on the destructive activity of ROS and RNS which leads to cell damage by the uncontrolled oxidation of proteins, nucleic acids, and lipid compounds or to programmed cell death (PCD). Conversely, the regulatory role of hydrogen peroxide (H_2_O_2_) or nitric oxide (NO), acting as signaling molecules in plant development, cell cycle progression and/or photosynthetic efficiency modulation, has also been demonstrated [1,4]. Data concerning the role of H_2_O_2_ and NO in the regulation of cell cycle of unicellular green algae are limited, but form a basis for the emplacement of ROS and RNS as important elements that may influence the course of algal cell cycle progression [5].

## 2. ROS and RNS in Plant and Algal Cells

The induction of oxidative stress is believed to be a universal response of plant and algal cells to adverse environmental conditions. Oxidative stress refers to the uncontrolled production of ROS, such as superoxide anion (O^2.-^), H_2_O_2_, hydroxyl radical (OH), and singlet oxygen (^1^O_2_). Among ROS, H_2_O_2_ has received the most attention as a signaling molecule [6]. As compared to the other ROS, it is relatively stable and less reactive. These intrinsic chemical properties, its synthesis and accumulation in several subcellular locations, its potential mobility within and between cells, and its rapid decomposition by enzymatic antioxidant systems make H_2_O_2_ an ideal signaling molecule [7]. The overall redox state of a cell was found to be a central hub in the regulation of many biochemical processes, and the H_2_O_2_ flux between cytosol, chloroplast, and mitochondrion is thought to be one of the regulatory mechanisms affecting cell growth via photosynthesis and respiration regulation [8]. Moreover, H_2_O_2_ seems to play a role in the regulation of gene expression encoding for antioxidant enzymes such as superoxide dismutase (SOD), ascorbic peroxidase (APX), catalase (CAT) and other defense proteins, for example, pathogenesis-related proteins (PRP) or heat-shock proteins (HSP) [9] (The role of H_2_O_2_ in chloroplast division [10] and chloroplast–mitochondria cross-talk, both on protein activity and genes expression levels, has also been demonstrated [11]). The detailed regulatory mechanism(s) of genes’ expression by intracellular H_2_O_2_ remains the subject of intense investigation. The available experimental data indicate the direct interaction of H_2_O_2_ with some receptor proteins [12], its interaction with redox-sensitive transcriptional factors (e.g., NPR1 and HSFs) [13], or direct inhibition of phosphatase activity leading to modification of signal transduction cascades [14,15]. One of the widely discussed ROS and/or RNS regulative actions is their interaction with the cysteine residues, leading to post-transcriptional modification (PTMs) of proteins [16,17]. Reversible PTMs include the formation of disulfides via reaction with low molecular weight thiols such as glutathione (S-glutathionylation) (-SSG), reactions with ROS to generate S-sulfenic acid (-SOH), or reactions with RNS to undergo S-nitrosylation (-SNO) [17].

The short-lived, gaseous molecule NO can be either enzymatically or non-enzymatically synthesized in plant cells [18,19]. NO production is mainly achieved via an oxidative route through nitric oxide synthase (NOS) activity that oxidizes L-arginine into L-citrulline and NO, or through the reduction in nitrite by nitrate reductase (NR). This latter process could involve the interaction of NR with NO-forming nitrite reductase (NOFNiR) [20]. It was found that NO plays a role in the regulation of various biochemical and physiological processes, including stress responses [21,22,23], developmental processes (leaf expansion and root growth), and phytoalexin production [24]. Furthermore, some data suggest the influence of NO on plant cell division [25]. In algal cells, NO involvement in cell cycle regulation was considered, although there is scarce data to support this claim. For example, a coincidental NO level and cell cycle progression and pattern was suggested for *Chlamydomonas reinhardtii* [26], and NO peak was recorded for *Chlorella* (Chlorophyta) species during population transition from lag phase to exponential growth phase [27]. The presence of NO, in the range of 10^−8^–10^−9^ mol/L, in the culture medium of several microalgae, such as *Heterosigma akashiwo* (Ochrophyta, Raphidophyceae), *Chaetoceros curvisetus*, *Skeletonema costatum* (Bacillariophyta), *Tetraselmis subcordiformis* (Chlorophyta), and *Gymnodinium* sp. (Miozoa) has also been reported [28,29]. The effects of NO on these algal growth patterns was strictly related to ambient conditions, such as trace elements composition, light, temperature and salinity [28,29]. The role of NO was also investigated in the seaweed *Gracilaria chilensis* (Rhodophyta), where carbon and nitrogen assimilation is found to depend on NO level [30]. Based on these findings, NO is now considered as a gaseous messenger in microalgae and one of the probable controlling factors of growth.

At the molecular level, NO signaling mainly relies on NO-dependent PTMs, with S-nitrosylation being the only demonstrated mechanism [31]. The occurrence and importance of tyrosine nitration and metal-nitrosylation, another NO-dependent PTM [31], remain to be addressed. Conversely, the possibility that the NO signal in algae could be transduced through an NO–cGMP signaling cascade has also been proposed [32]. The existence of soluble guanylate cyclases (sGC) activated by NO has been documented in *Chlamydomonas reinhardtii*, although the downstream effectors including cGMP-dependent protein kinases (PKG), cyclic nucleotide-gated channels (CNGC), and phosphodiesterases (PDE) have not been reported [33]. The turnover of NO signaling relies on the regulation of the S-nitrosylated protein pool by Trx and GSNOR, the later controlling the content of the S-nitrosylating agent GSNO.

## 3. H_2_O_2_ and NO Cross-Talk

Many cellular processes are controlled by the ROS and RNS redox balance because local changes in the redox environment mediate the spatio-temporal regulation of protein functions and enzyme activities in a compartment-specific manner [34]. The role of ROS and RNS cross-talk has been discussed in relation to the higher plants but is nearly unexplored with regards algal cells and their development. Exogenous H_2_O_2_ was observed to induce NO generation, accompanied by a substantial increase in an NOS-like activity by H_2_O_2_ treatment [35]. For example, in *Arabidopsis thaliana*, cells synthesize NO in an interdependent manner with H_2_O_2_ production, highlighting that, in *Arabidopsis thaliana*, abscisic acid (ABA)-mediated NO generation is in fact dependent on ABA-induced H_2_O_2_ production. Using the guard cell model system, stomatal closure induced by H_2_O_2_ was found to be inhibited by the removal of NO with an NO scavenger, and both ABA and H_2_O_2_ stimulate guard cells to produce NO [36]. Although many reports suggest that NO and H_2_O_2_ can regulate each other’s synthesis, some inconsistencies still exist. Synthesis of NO is stimulated only directly after the application of exogenous H_2_O_2_, but not after prolonged exposure to H_2_O_2_, supporting the view that distinct signaling mechanisms are activated under specific combinations of several factors [37]. Furthermore, it has been shown that H_2_O_2_ is able to induce the activation of mitogen-activated kinase (MAPK) in leaves, cell cultures, and protoplasts of *Pisum sativum* L. [34]. Comparatively, the NO donor, sodium nitroprusside (SNP), can also induce the transient activation of a similar kinase [38]. The above data suggest that H_2_O_2_ and NO may converge on MAPK signaling pathways in regulating cell functions. Another possible means of NO signaling are the cyclic nucleotide GMP (cGMP)-mediated pathways. Although this has so far only been described in animal cells [23]). There are indications that cGMP is also synthesized in plant cells, and this synthesis is enhanced by NO [39].

There is a scarcity of data [40] indicating that H_2_O_2_ and NO can modify algal cell development by influencing the expression of cell cycle-related proteins, such as cyclin and cyclin-dependent kinases and histone proteins [5,41]. These examples form a basis for establishing the role of ROS and RNS in the cell cycle of green algae, both as the signals directly involved in the regulation of protein expression required for completing the cell cycle, and as factors involved in the regulation of photoautotrophic growth [42], which, in turn, allow the cell to attain the commitment competence. Carvalho et al. [43] also proposed the occurrence of the circadian, day–night cycle-related regulation of algal cell redox states, in which the circadian oscillations in the expression of enzymes directly involved in H_2_O_2_ metabolism (SOD, PXs and CAT) are highlighted as one of the main factors determining the H_2_O_2_ content in the cell. The complementary nature of these studies was a complex analysis of changes in the transcriptome profile of synchronously grown *Chlamydomonas* presented by Zones et al. [44]. The authors characterized genome-wide diurnal gene expression and stated that over 80% of the measured transcriptome was expressed with strong periodicity. The subgroups of a transiently expressed cluster of light-stress-related genes were identified, where the genes responsible for H_2_O_2_ and NO metabolism were present [44]. In the natural environment, the day and night periods are connected with photosynthesis- and respiration-related ROS generation. These induced the occurrence of an extreme case of growth optimization for unicellular organisms exposed to light/dark cycles of different lengths, namely division by multiple fission. Multiple fission provides a large advantage to cells since they can grow for the entire light phase and then nuclear DNA replication–division sequence is performed in the dark. The clear benefit of such a development pattern is ‘not wasting’ time for division when growth is possible. Secondly, DNA replication is protected from exposition to potentially mutagenic UV irradiation, a component of natural day light [45].

## 4. The Cell Cycle of Unicellular Green Algae

The cell cycle of many green algae species differs from that which occurs in higher plants. The typical course of a plant cell cycle is similar to other eukaryotic organisms and entails a growth phase, synthesis phase, second growth phase, followed by mitosis and cell division. This type of green algal cell cycle, where mother cells divide into two progeny cells, is known as binary fission and occurs in slow-growing species such as *Ostreococcus tauri* (Chlorophyta) [46] which are unable to achieve more than one doubling of biomass during a single light period. When the fast-multiplying algal species are considered, this basic rule “one mother cell = two daughter cells” is always maintained, although the overlapping of growth and division processes that occur during one cell cycle results in one mother cell giving rise to more than two daughter cells, meaning that they divide by multiple fission. In this type of cell multiplication, one mother cell divides into 2^n^ cells, where “n” is an integer from 1 up to 10. The number of daughter cells is dictated by external conditions. However, the maximum number of daughter cells that can be produced is fixed in different species and usually ranges from 8 to 16 (i.e., n = 3–4). A good example of green algae that divide by multiple fission is a model organism *Chlamydomonas reinhardtii*. However, the historical research on the progress of multiple fission algal cell cycle concerns *Chlorella* [47] (Tamiya, Shizuoka, Japan, 1966) and *Desmodesmus* (formerly *Scenedesmus*) genera [48,49,50]. Interestingly, the studies performed on *Chlamydomonas* indicate the involvement of many redox-dependent transcription factors (TF) analyzed both on the enzymatic and genes expression level, facilitated by advances in genome sequence analysis [51].

The green alga *Chlamydomonas* (*Chlamydomonas reinhardtii*) with its numerous mutants has been a model organism to study biological processes such as photosynthesis, respiration, and phototaxis, among others [52]. *Chlamydomonas* has metabolic and molecular pathways that are both analogous to those in the model plant *Arabidopsis* (*Arabidopsis thaliana*), together with some that are distinct, offering a glimpse into the diversity of photosynthetic organisms. *Chlamydomonas* has several advantages as an experimental organism for studying redox regulation and ROS signaling [53]. Because it is unicellular, *Chlamydomonas* can be grown in uniform cultures, and ROS induction and treatment are relatively easily compared with multicellular plants. The growth of *Chlamydomonas* cells in culture may be easily synchronized via natural factors such as light/dark cycles without using artificial chemical compounds. Conversely, the overlapping rounds or replications can obscure the view of events of one replication round, although it may also be used as a tool to study multiple fission cell cycles where modification of light/dark duration allows one, two, three, or more replication rounds in the course of the cycle.

The cell cycles in cells dividing by multiple fission are characterized by relatively long growth phase (G_1_) and an almost complete absence of G_2_ phase that results from the synthesis of structural proteins, karyokinetic spindle and DNA replication that are overlapping [54]. At the end of G_1_, the cells are competent to enter the subsequent cell cycle phases, i.e., individual replicative processes. The G_1_/S-phase transition undergoes regulatory mechanisms at the cell cycle checkpoint [55]. Cell size is a critical point (“sizer”) for the G_1_/S-phase transition, but the G_1_ phase duration (“timer”) should not be neglected either [56]. The checkpoint at the end of G_1_ phase is called the commitment point and is assumed to be equivalent to the “Start” point in yeast cells [57] or “restriction” point in mammalian cells [58]. After passing this point, the cell cycle could be completed in the dark without a supply of external energy [54]. The replicative processes are restricted to the end of the cell cycle. Therefore, the cells remain mononuclear, and the amount of DNA is constant for most of the cycle.

*Chlamydomonas* encodes orthologs of the major plant CDK and cyclin families [59]. Furthermore, *Chlamydomonas* also encodes two CDK subtypes and two cyclin subtypes that are not found in higher plants, fungi, or animals. Besides cyclins and CDKs, orthologs of a negative regulator of CDKs, CKS1/suc1, a CDK subunit, and potential CDC25 homologs and E2F and DP orthologs have also been identified in *Chlamydomonas* [59]. *Chlamydomonas* encodes a single ortholog for each of the plant-type CDKs A, B, C, D, and E (genes were designated *CDKA1*, *B1*, *C1*, *D1*, and *E1*, respectively) but does not encode an F-type CDK. In addition, *Chlamydomonas* encodes four proteins of the CDK family that are not orthologous to any known CDKs in plants or animals. Two of these CDKs are related to each other and encoded by genes designated *CDKG1* and *CDKG2*, but are significantly diverged in their predicted C-helices. mRNA for *CDKA1* is present constitutively during the cell cycle with expression increasing as cells enter the growth phase at the beginning of the light period and increase further around the time of the S/M phase. mRNA for *CDKB1* shows two peaks of expression, one corresponding to the passage through the commitment phase and a second, very strong peak during the S/M phase [59]. Elevated expression of the CDKB subfamily proteins is consistent with a role for CDKB in the regulation of the G2/M phase transition, together with a single *Chlamydomonas* ortholog for CYCA1 and CYCB1 cyclins, and three D-type cyclins, CYCD1, CYCD2, and CYCD3. Interestingly, both CYCA1 and CYCB1 have a destruction box (D-box) in their N-terminal domains. The expression profiles of both *CYCA1* and *CYCB1* resemble *CDKB1* with elevated levels of mRNA appearing at commitment and during the S/M phase. Expression of a subset of *Chlamydomonas* genes (e.g., *CYCA1*, *CYCB1*, *CDKB1*, *RNR1*, and *POLA4*), most of which are likely to be RB-E2F pathway targets, is up-regulated prior to commitment and down-regulated just after passage through it [60]. These same genes are up-regulated a second time during the S/M phase, which is when they are likely to be functional. Unlike higher plants and animals, most of the core cell cycle regulatory genes in *Chlamydomonas* are present only in single copy.

One feature of *Chlamydomonas* is the occurrence of only one chloroplast per cell [61]. Thus, the division of this organelle requires coordination with the whole cell division process, with respect to the existence of the overlapping sequences of nucleic acid replication followed by mitosis. In algae, the chloroplast divides once per cell cycle before the host cell completes cytokinesis [62]. In the cells of *C**. reinhardtii* as well as in the cells of higher plants, the redox state of the cell, overall regulated by the reduced glutathione pool, was found to be a key regulator of the DNA replication [63,64,65]. For example, in *Arabidopsis thaliana* GSH-deficient root meristemless 1-1 (rml1-1) mutant, GSH depletion was found to increase significantly the redox potentials of the nucleus and cytosol, and to arrest the cell cycle in roots [64]. In the work describing *Chlamydomonas reinhardtii* smt15-1 single mutants (*suppressor of mating type locus3 15-1 mutant*), exhibiting a cell cycle defect leading to a small-cell phenotype, an increase in total glutathione levels was shown. What is more, mutants’ cells exhibited an increased reduced-to-oxidized glutathione redox ratio throughout the cell cycle [65].

DNA replication was found to be regulated independently from chloroplast division, where the redox state was sensed by the nucleoids, and the disulfide bonds in nucleoid-associated proteins were involved in this regulatory activity [66]. These results show that chloroplast DNA replication occurs independently of either the cell cycle, or the timing of chloroplast division. Furthermore, chloroplast DNA replication occurs when light is available in photoautotrophic culture, but also during darkness when cells are grown heterotrophically. It was demonstrated that chloroplast DNA replication in vitro was activated by DTT, known as a reducing agent, but it was inactivated by diamide, which is a sulfhydryl group-specific oxidative agent [66]. These results highlighted the activity of the chloroplast DNA polymerase as being regulated by the redox state of the sulfhydryl group in certain chloroplast nucleoid-associated proteins. The chloroplast nucleoids are sufficient to sense the redox state to change the DNA replication activity.

## 5. Redox-Dependent Components of Cell Cycle Machinery

Internally (e.g., circadian clock-related), as well as externally (e.g., biotic and abiotic stressors) induced oxidative and nitrosative signals are important components of cell cycle regulation, leading to either promotion or arrest [66,67,68], and this relation seems to be universal among plants and animals [69,70]. In mammalian cells, cell cycle progression is driven by an intrinsic redox cycle [71]. Internal and external growth stimuli activate the cyclin D–CDK complex, which phosphorylates retinoblastoma (RB), leading to the release of the transcription factor E2F (elongation 2 factor) and enabling the G1/S transition. The binding of growth factors, e.g., epidermal growth factor (EGF), to dedicated receptors (such as EGFR) can be promoted by oxidation mediated via mild ROS accumulation. Moreover, redox-dependent control of a eukaryotic nuclear RNA polymerase, followed by regulation of the transcription of genes encoding essential RNAs, may contribute to ROS-induced mammalian cell proliferation program variation.

The concept of redox regulation of plant cell development, with special consideration to proteins involved in the cell cycle, is also reported in the literature [72,73]. Oxidative perturbations were found to be a trigger that led to changes in the levels of CYC and CDK transcripts and their activities [74,75]. Proteins from the CDK A subfamily in plants regulate the G1/S and G2/M phase transitions and play a role during the S phase, while CDK B proteins act on the G2/M transition and during the M phase [76]. With regards to cyclins, the D-type CYCs (CYCD) function jointly with CDKA during the G1/S transition, when A-type CYCs (CYCA) operate during the S phase, and together with B-type CYCs (CYCB) to regulate the G2/M transition. The CYC B1 and CDK1, during the G2-to-M transition, may participate in the mitochondria complex I subunit phosphorylation, enhancing mitochondrial respiration and ATP production to run cell cycle progression [77]. In plant cells, activation of CDKs requires phosphorylation by CDK-activating kinases (CAKs). In addition, ubiquitin ligases such as the anaphase-promoting complex/cyclosome (APC/C) and Skp1/Cullin/F-box protein (SCF)-related complex act to remove cell cycle regulators by proteolysis. Many genes expressed at the G1/S-G2/M phases in plants are regulated by the transcription factors E2F and MYB3R, members of a multiprotein RBR–MYB3R–E2F complexes, believed to be homologous to the DREAM complex in animals [78]. RBR (retinoblastoma related protein, mammalian RB relative) is a good example of an ROS-regulated element of cell cycle that has been thoroughly investigated in both animal and higher plant systems. In plants, it promotes phosphorylation induced by mitogenic signals, through the action of CDKs in association with D-type cyclins, particularly CYCLIN D3:1 (CYCD3:1) [78]. In *Arabidopsis*, a RBR1 is a conserved regulator of cell proliferation, differentiation, and stem cell niche maintenance [79]. RBR1, as a signal-dependent scaffold protein, is regulated by phosphorylation-induced conformational changes that support a range of interactive surfaces for diverse complexes and functions [80,81]. RB regulates cell proliferation by decreasing the E2F-dependent transcription of cell cycle genes. Before phosphorylation, the E2FB transcription factors are released from RBR, which enhance cell cycle gene expression and stimulate cellular proliferation. In animals, the hypo-phosphorylated forms of RB bind to E2F transcription factors during G1, leading to the inhibition of cell cycle by E2F-mediated gene expression [82]. Moreover, RBR represents diverse functions through phosphorylation during the early G1 phase [83]. During G1, RBR is phosphorylated on alternative individual sites, leading to the formation of mono-phosphorylated RBR, which allows it to associate with E2F1 to control arrest checkpoints and apoptosis (e.g., if DNA damage occurs), or the dissociation from E2F4, thus enabling cell entry into G1/S. Additionally, the nutrient-dependent life-cycle transitions of *Chlamydomonas* regulatory component known as DREAM (evolutionarily conserved multi-protein transcriptional regulatory complex: DP, RB, E2F and Myb-MuvB may also play a role in redox-dependent cell cycle regulation [84]. The regulatory role of RBR in plant cell proliferation can be separated from its novel function in safeguarding genome integrity [85]. The RB protein also shows E2F-independent functions through binding to other nuclear or extra-nuclear partners. So far, we can speculate that redox-related modification of proteins, such as RBR in plants, may provide a similar, and likely even more important, level of control to the cell cycle, in a manner similar to that observed in animals.

However, contrary to animals and higher plants, the direct involvement of ROS and RNS in cell cycle progress-related gene expression in algal cells is devoid of investigation, besides a few reports [5,86]. However, the highly evolutionarily conserved level of cell cycle machinery genes among eucaryotic organisms [87] suggests that the evidence obtained for higher plants can be universal, and applied to algal cells, since most of the cell cycle components of higher plant cell cycle machinery have been identified in the *Chlamydomonas* genome [51,59,88]. All these findings support the concept tha“ "oxidative stress”-sensitive checkpoints are important in the regulation of the cell cycle [68,89]. The complex control of the cell cycle via “overall cell redox state”, often explained simply in terms of a given threshold ROS level, as well as their relative proportions, is a factor required to drive cells into proliferation or cell cycle arrest [90]. The outcomes of cellular oxidative signaling pathways depend on a number of parameters, including the chemical nature of the ROS form that is produced (i.e., superoxide, hydrogen peroxide or singlet oxygen) and the nature of their interacting partner (protein thiol, metabolite, lipid or DNA molecule), together with cell identity. Table 1 summarizes the proteins involved in cell cycle progression, whose expression was found to be regulated by the ROS and RNS intracellular changes in *Arabidopsis thaliana* [91] with their homologs found in the *Chlamydomonas* genome. These regulations were indicated both by the direct interaction with signaling molecules on the redox-sensitive transcription factors, and also the induction or arrest of signaling pathways including MAPK, G-protein and calmodulin-dependent pathway. Most of them possessed cysteine residues, which were identified or predicted by e-data estimation and which potentially allowed the ROS/RNS to mediate conformational changes [91]. Structural and functional analogues of most of these genes can be found in the *Chlamydomonas* genome, according to JGI Phytozome13 Plant genomic resource (Table 1).

## 6. ROS and RNS Interactions with Photosynthesis

In photoautotrophic organisms, photosynthesis is the primary source of energy that drives cells growth. Thus, the ability of cells to complete cell division, through the strict set of parameters and cell signaling, depends on the efficiency of photosynthetic reactions [92]. During this time, the chloroplast is the main source of ROS, and the photosynthetic activity is an important component of NO generation via NR and NiR pathway [23]. Thus, the participation of the ROS and RNS in cell cycle progression occurs in the coordination of chloroplast photosynthetic activity and division of this organelle, together with cell division coordination [93]. Such involvement of the redox state was evidenced in the coordination of chloroplast and nuclei DNA replication, light harvesting complex formation and operation, together with expression/activity regulation of most of the Calvin–Benson cycle. Chloroplast thylakoid membranes contain a well-established system for redox control of the post-translational modification of a few proteins [94], including the major light-harvesting complex, LHC II [95]. LHC II phosphorylation activation by protein kinase occurs when the pool of plastoquinone between the two photosynthetic (PS) reaction centers-I and II-becomes reduced to plastoquinol. This allows LHC II to change its destination from photosystem II to photosystem I. As the plastoquinone pool is reduced by photosystem II and oxidized by PSI, the reduction in plastoquinone is the trigger, which initiates the PS apparatus response to adverse conditions. In similar context, it can be expected that the cytochrome b6/f complex can play a significant role in the redox status-mediated gene expression. Its location in the midway between PSI and II, at a strategic point in the electron transfer chain, and its ability to participate in both linear and cyclic electron transport, make it a redox-status-sensing hub in photosynthetic electron transport [53]. During the cyclic electron transport, ATP synthesis progresses independently of the reduction in the terminal electron acceptor NADP+, and the cytochrome b6/f complex is both oxidized and reduced by PS I. A redox sensor located in or near the cytochrome b6/f complex could help regulate the proportion of electrons recycled into the cytochrome b6/f complex from the acceptor side of PSI, and hence may help regulate the production of ATP relative to NADPH [53]. Cytochrome b/c complexes contain quinone-binding sites and respond directly to changes in the redox states of quinone pools which may enable them to be redox sensors [53].

Most species investigated so far, including unicellular model green alga *Chlamydomonas*, exhibit divergent expression changes in genes encoding the reaction-center of PSI and PSII components. Upon reduction in the PQ pool, expression of PSI-related genes is promoted, while upon its oxidation, expression of PSII-related genes rises. This relation also involves the action of the STN7 kinase, that catalyzes the phosphorylation of membrane proteins of LHCII in chloroplasts [96]. Within plastids, redox signals must be transduced only for a short distance to the plastome [97,98,99]. The long-term response (LTR) controlling chloroplast genes for photosynthetic core proteins involves the action of the thylakoid membrane kinase STN7 [96]. This kinase might be a component for PQ redox signal sensing and transducing via a (still putative) phosphorylation cascade acting on gene expression machinery. Such a model of a phosphorylation cascade is in a line with earlier models that describe phosphorylation of sigma factors of the RNA polymerase as the controller of plastid transcription [100,101]. The involvement of thiol groups in redox potential sensing is widely discussed in the general context of plant cell biology. Redox control of plastid gene expression may involve thioredoxin regulation of thiol groups in the disulfide isomerase protein. Thioredoxins can move easily through the stroma, and provide an easy mode of signal transduction. The model has been proposed in *Chlamydomonas reinhardtii*, but some data suggest that it might also be valid for higher plants [102]. In this way, control is exerted on the binding of an RNA-binding complex to psbA mRNA and the subsequent initiation of translation [103]. Since redox-dependent signal transduction in chloroplasts and their involvement in the regulation of expression of chloroplast-encoded genes is based on relatively short distance interactions, transduction of photosynthetic redox signals that control nuclear gene expression appears to be more complex than that within plastids. These signals need to leave the plastid, pass the cytosol and enter the nucleus before triggering a dedicated response [104]. Two main types of signal-transduction mechanisms have been proposed. In the first, the redox signal is sensed by an internal plastid system and transduced over the envelope, whilst in the second, a redox-regulated compound can directly leave the plastid. In this context, it is assumed that H_2_O_2_ can freely diffuse across the chloroplast envelope and that it activates a MAPK cascade in the cytosol, affecting nuclear gene expression [105]. Photosynthetic redox signals generated in the electron transport chain are associated directly with many environmental factors, and indirectly with the efficiency of the Calvin–Benson cycle CBC [106]. Proteomic approaches indicated almost all Calvin–Benson cycle enzymes to be susceptible for carbonylation (ROS mediated) and/or nitrosylation (RNS mediated) modifications. These regulations have not been confirmed biochemically with the exception of *Chlamydomonas*’ chloroplastic triosephosphate isomerase (TPI) which was shown to be partially inhibited by nitrosylation [107] and for the land plant Rubisco, in which it appears to be inhibited by nitrosylation [108]. Recent proteomic studies allowed identification of the cysteine residues undergoing nitrosylation and glutathionylation and forming a TRX-reducible disulfide bond. Many of these cysteines are conserved, especially in photosynthetic organisms. [4]. All these data suggest that Calvin–Benson cycle enzymes are regulated by a complex and highly dynamic network of redox PTMs. This knowledge is advantageous considering the principal role of the Calvin–Benson cycle in the determination of plant adaptation to stress conditions, CO_2_ fixation and biomass production. Since the growth rate of algal cells is one of the highest among photosynthetic organisms, the role of ROS and RNS may occur to be one of the key components affecting algal cell growth and achieving of commitment competence, determined by “sizer” mechanism.

## 7. ROS and RNS Mediated Control of Cell Cycle-Evidence in Higher Plants and Algae

### 7.1. Case 1: Medicago Sativa (Alfalfa)

The involvement of ROS in cell cycle activation (G0-to-G1 transition) of plants was initially presented by Fehér et al. [82]. The authors provided evidence that oxidative stress plays a role in CDKA1 activation during the re-entry of cultured Alfalfa cells into the cell cycle. Alfalfa leaf cells and freshly prepared protoplasts constitute a reliably homogenous population of mesophyll cells, most of which were at the G0-phase of the cell cycle. Up to the moment of protoplast isolation, these cells were specialized for photosynthesis, with RUBISCO (ribulose-1,5-bisphosphate carboxylase/oxygenase) comprising more than 70% of their protein content. In this state, most nuclear genes were switched off in these cells, and their nuclei were characterized by being compact with highly condensed chromatin and a small nucleolus. Cultured alfalfa cells in stationary phase (11 days without subculture) suspension were transferred to fresh media and the cell cycle re-entry was monitored in the absence and presence of 30 µM CuSO_4_, as an inducer of mild oxidative stress, and 1 µM DPI (diphenyleneiodonium) as NADPH oxidase inhibitor. This treatment transiently enhanced the CDK activity between 24–48 h after subculture that was prevented by parallel DPI application. DPI alone resulted in only a slight decrease in CDK activity but still inhibited cell division as was indicated by the decreased growth rate of the cultures. A limiting level of exogenous auxin, together with externally induced moderate oxidative stress (provoked, e.g., by 50 µM CuSO_4_) promoted G0-to-G1 transition as indicated by the increased cell size, changes to the chromatin structure and RNA-dependent fluorescence of nuclei/nucleoli.

### 7.2. Case 2: Arabidopsis

In 2017, De Simone et al. [109] characterized redox changes in the nuclei and cytosol during the mitotic cell cycle in the embryonic roots of germinating *Arabidopsis* seedlings. These data demonstrate the presence of a redox cycle within the plant cell cycle and indicate the redox state of the nuclei as an important factor in cell cycle progression. Controlled oxidation was found to be a key feature of the early stages of the plant cell cycle. On the other hand, sustained mild oxidation restricted nuclear functions and impairs progression through the cell cycle leading to a smaller number of cells in the *Arabidopsis* root apical meristem. Data presented in this paper provide the evidence that transient oxidation occurs at G1 in the cytosol and nuclei of proliferating cells in the embryonic root, showing that redox cycling occurs within the plant cell cycle as it does in animal cells. Moreover, the evidence presented here clearly demonstrates that transient oxidation of the cytosol occurs early in the G1 phase of the cell cycle and precedes transient oxidation of the nuclei. The high level of oxidation of nuclei in roots with low antioxidant buffering capability demonstrates that a change in the nuclear redox state of about 20 mV is sufficient to delay G1 phase and subsequent progression through the cell cycle. This finding is consistent with previous observations showing that mild oxidative stress experienced during the S phase was able to delay entry into mitosis because the cells had a longer G2 phase. According to these data, oxidation (high level)-dependent cell cycle arrest is generally associated with an inhibition of the activity of CDKs, cell cycle gene expression, and concomitant activation of stress genes.

### 7.3. Case 3 and 4: Chlamydomonas

Transcriptomic analysis of *Chlamydomonas* genes, presented by Blaby et al. [88], provided an overview of the impact of H_2_O_2_ on protein stress responses and its overlap with other stress related transcriptomes. Among this, data concerning oxidative stress occurrence during the *Chlamydomonas* cell cycle was presented. The transcriptome experiment identified a ‘light-stress cluster’ of 280 genes that was transiently upregulated at the onset of the light phase but remained near-undetectable across remaining time points in the dark phase, similar to ones required for cell synchronous culture maintenance. Ninety-nine of these genes were significantly differentially abundant in response to H_2_O_2_ (56 increased mRNA abundance, 43 decreased between 0 and 1 h following H_2_O_2_ treatment). For transcripts with a general function, the 1 h overlap contained transcripts involved in the metabolism of DNA and/or RNA and proteins folding. Ultimately, an increasing enrichment of protein metabolism-related transcripts occurred. The authors found a significant number of transcripts induced by H_2_O_2_ with an abundance peak after the transition from dark to light, and as cells mature during the day. Enrichment within the light stress cluster at the beginning of the day and towards the end of the day was shared with transcripts induced by singlet oxygen. At the 22 h time point, there was an enrichment of transcripts related to ROS detoxification/regulation. *Chlamydomonas* cells experience an extended G1 phase during the day, during which biomass is accumulated. Moreover, cell division was arrested until the night. Based on the functional classification of the corresponding proteins, the H_2_O_2_-induced transcripts with maximum abundance at the end of the day largely encode proteins involved in protein turnover. Approximately 2 h prior to the transition to light, a spike in ROS-induced transcripts, possibly highlighting clock-regulated transcripts, are maximally abundant in anticipation of the transition to the light period. Although there was a significant enrichment of H_2_O_2_-induced transcripts with peak abundance at this time point, we observed a higher significance associated with RB-induced transcripts, possibly highlighting anticipation of photo-oxidative damage associated with the shock of switching from dark to light. The abundance of the *Chlamydomonas* ortholog of *Arabidopsis* CCA1 (Cre06.g275350; circadian clock-associated 1), which has been demonstrated to regulate the induction of oxidative stress responses in that organism [110], was slightly reduced throughout our time course. In the *Chlamydomonas* diurnal experiment, H_2_O_2_ mediated peak abundance occurs immediately prior to the light period, suggesting that Cre06.g275350 is regulated by cell cycle-related circadian rhythms.

Using the synchronous cultures of *Chlamydomonas reinhardtii*, Pokora et al. [5,26] demonstrated the coincidence of the changes in the concentration of H_2_O_2_ and NO with the time-lapse of the key checkpoints of the *Chlamydomonas* cell cycle, such as G1/S and S/M transitions, or progeny cell release [26]. These changes were found to be a result of cell cycle-related circadian oscillation in the expression of enzymes responsible in *Chlamydomonas* cells for H_2_O_2_ and NO production and scavenging (SOD, CAT, APX, NR). It was hypothesized that changes in concentrations of H_2_O_2_ and NO, when induced externally, should lead to a modification of the life cycle of microalgae by modulation of the expression of genes required for cell cycle progression. To verify this theory, a shift in NO/H_2_O_2_ ratio through exogenous application of H_2_O_2_ to the algal suspension at selected time points in the *Chlamydomonas* cell cycle was investigated. H_2_O_2_ was applied exogenously to the culture to reach concentrations 1.5 times higher than those indicated in control cells at the time of application. These external H_2_O_2_ applications modified the course of the cell cycle but did not cause visible signs of toxic effects on cells (Figure 1).

Depending on the phase of the *Chlamydomonas* cell cycle, externally provided H_2_O_2_ had (1) accelerated or delayed the duration of the cell cycle, (2) increased the number of replication rounds occurring in one cell cycle, and (3) modified the biomass and cell volume of progeny cells, or accelerated the liberation of daughter cells. From the above, two experimental variants seem to be most interesting: when H_2_O_2_ was applied during the 3rd or 6th h of the light period of the cell cycle.

Hydrogen peroxide, when applied in the 3rd h of the light period of *Chlamydomonas* synchronous cultures exhibited a stimulatory effect on photosynthesis efficiency in the following hours. Chlorophyll a fluorescence analysis indicated that energy absorption and trapping by a single active PS II reaction center remained unchanged, but its utilization in the electron transport chain was markedly higher. Within the following 5 h (3rd to 8th h of the cell cycle), the efficiency of electron transport was nearly 50% higher than in the control cells, and this enhanced photosynthesis resulted in the increased cell volume. This more-efficient growth initiated before the “commitment sizer” control point, usually placed at about the 6th h of light-dependent cell growth [54], led to the occurrence of an additional fourth round of replication. It was evidenced by the higher amount of total DNA in the hour corresponding to the S phase of the cell cycle. Moreover, transcripts abundance of CYCs and CDKs were approximately 30% higher and they typical expression pattern, occurred 1 h earlier, than those in the untreated cells. The total number of daughter cells released at the end of the dark phase was also around 30% higher, which suggests that about half the population undergoes an additional round of replication, leading to the release of 16 daughter cells from 1 mother cell.

In the same experiment, the application of H_2_O_2_ in the 6th h of the light period of the *Chlamydomonas* cell cycle resulted in the delay of the further cell cycle steps. Interestingly, the final number of released daughter cells was similar to the control, but the average volume and biomass of the single zoospore were higher. In the *Chlamydomonas* cell cycle, division of chloroplasts occurs just before cytokinesis and is associated with a marked decrease in photosynthetic activity [54,62]. Analysis of CYC and CDKs expression profiles and PS II activity of the control cells suggests that chloroplast division should occur in the last hour of the light phase cell cycle [26]. In the cells treated with H_2_O_2_ in the 6th h of culture, 1 h delay in the course of CYC and CDKs expression occurred, along with DNA synthesis. At the same time, the energy utilization efficiency in the electron transport chain in the last two hours of cells’ light-dependent growth was higher than in the control culture. Under such conditions, the increase in energy was probably not sufficient to enable *Chlamydomonas* cells to pass an additional “mitotic sizer” checkpoint. Thus, cells were similar to the control culture at the occurrence of three replication rounds in one cell cycle, but due to the extended photosynthetic activity, produced larger zoospores.

## 8. Summary

The regulative role of ROS and RNS, such as H_2_O_2_ and NO, has been thoroughly investigated in higher plants, although less is known with regards to algae. Both H_2_O_2_ and NO can act as signaling molecules in plant development, cell cycle progression and/or photosynthesis efficiency modulation [1,4]. Data concerning the role of H_2_O_2_ and NO in the regulation of the cell cycles of unicellular green algae are limited but form the basis for the emplacement of ROS and RNS as important elements that may influence the course of algal cell cycles. In the model organism *Chlamydomonas reinhardtii*, it was shown that the expression of nearly 300 genes is under the control of H_2_O_2_, where most of them are related to light response [88]. In the diurnal cycle, a significant number of these H_2_O_2_-induced transcripts, with peak transcript abundance just after the transition from dark to light and as cells grow during the day, was noted [88]. Moreover, exogenous application of H_2_O_2_ led to the modification of the reciprocal, quantitative H_2_O_2_/NO ratio followed by changes in the progression of the cell cycle which consequently affected microalgae cell growth. This manipulation may provide a useful tool to control the development of *Chlamydomonas* populations and obtain algal cultures with desirable features such as increased biomass and/or cell numbers. Externally induced, mild modification of redox homeostasis may accelerate or delay the cell cycle, increase the number of replication rounds, modify the volume of progeny cells, and accelerate daughter cell liberation. The effect on photosynthetic activity was also significant. The questions about the H_2_O_2_/NO balance effect on Calvin–Benson cycle enzyme activity and its implication to light-driven reaction as well as the detailed cellular level of redox signal detection still requires answers. Moreover, the long-time effect on the next generations of progeny cells, obtained from mother cells in which the cell cycle was affected, remains unknown. Further explanation of the basis of such phenomena will allow for an understanding of the regulatory role of H_2_O_2_ in the cell cycle and can be an important step for manipulating the development of microalgae in mass culture-based technologies.

## Figures and Tables

**Figure 1 cells-11-02425-f001:**
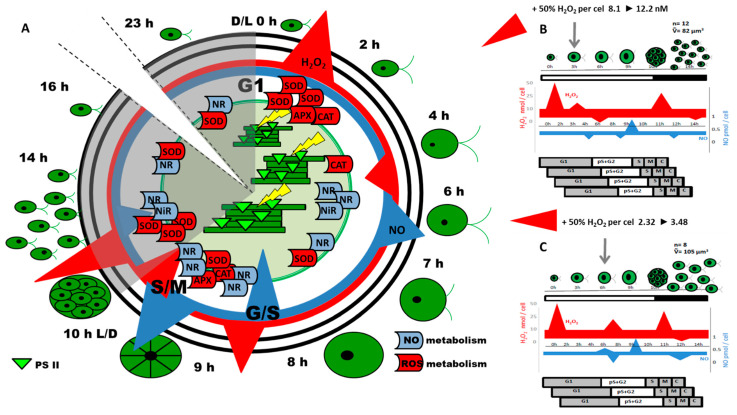
The overall plot of events during the cell cycle of *Chlamydomonas reinhardtii* under control conditions (**A**) and after exogenous application of H_2_O_2_ to a suspension of synchronously growing cells in the selected moments of the cell cycle: 3rd h of the light period (**B**) and 6th h of the light period (**C**). The time of H_2_O_2_ application to the culture is indicated by arrows. White areas indicate the light cell cycle period, while shaded ones represent the dark period. Changes in H_2_O_2_ and NO level during the light and dark periods are shown as triangles, where figures size and orientation represent the trends in the H_2_O_2_ and NO level changes during the cell cycle duration. Schematic pictures of the cells indicate their increasing size during the cell cycle and small circles with black dots inside illustrate the zoospores with cell nuclei arising within one mother cell. The average number (ñ) and the average size (V) of zoospores released from the mother cell in the dark phase of the cell cycle are indicated by numbers. Modified from [5,26]. Legend to Figure 1. APX–ascorbate peroxidase; CAT–catalase; D–darkness; L–light; NiR–nitrite reductase; NO–nitric oxide; NR–nitrate reductase; SOD–superoxide dismutase; G1,2, S, M, C–cell cycle phases: Growth, Synthesis, Mitosis, Cytokinesis.

**Table 1 cells-11-02425-t001:** Selected cell cycle-related genes, found to be potentially redox regulated; after Foyer et al., [91] (column 1 and 2); calculated by NetSurfP) and their homologues identified in the *Chlamydomonas reinhardtii* genome (columns 3–5), according to JGI Phytozome13 Plant genomic resource analysis (at least 65% homology). Abbreviation in 3rd column: n.d.–not defined for *Chlamydomonas*.

*Arabidopsis Thaiana*		*Chlamydomonas Reinhardtii*		
Name	AGI	Name	Locus ID	Note
ALY1	AT5G27610	REF1	Cre10.g462250	RNA Export Factor
APC6	AT1G78770	APC6	Cre13.g562950	Anaphase promoting complex subunit 6
ARLA1A	AT5G37680	ARL8	Cre17.g708250	ARF-like GTPase
ATM	AT3G48190	ATM1	Cre13.g564350	ATM-like serine protein kinase
ATMYB3R1/PC-MYB1	AT4G32730	n.d.	Cre09.g399067	–MYB-like DNA-binding protein MYB//ATMYB1 protein
ATXR5	AT5G09790	HLM3	Cre01.g041100	–Histone-lysine N-methyltransferase
CCS52A2	AT4G11920	CDH1	Cre09.g406851	Activator and specificity factor for anaphase promoting complex
CCS52B/FZR3	AT5G13840	CCS5	Cre17.g702150	Thioredoxin-like protein similar to Arabidopsis HCF164
CDC20.2	AT4G33260	CDKH1	Cre07.g355400	Cyclin-dependent kinase
CDC20.4	AT5G26900	CDKA1	Cre10.g465900	Cre10.g465900–Cyclin-dependent kinase
CDC45	AT3G25100	CDC45	Cre06.g270250	Cell Division Cycle protein 45
**CDC48**	AT3G09840	CDC48	Cre06.g269950	Cell Division Cycle protein 48
**CDC6**	AT2G29680	CDC6	Cre06.g292850	Pre-initiation complex, subunit CDC6
**CDKC;1**	AT4g28980	CDKC1	Cre08.g385850	Cyclin-dependent kinase
**CDKC;2**	AT5g10270	CDKC1	Cre08.g385850_4532	Cyclin-dependent kinase
**CDKD1;1**	AT1G73690	CDKD1	Cre09.g388000	CDK activating kinase
**CDKG1**	AT5G63370	CDKG1	Cre06.g271100	CDK activating kinase
**CDKG2**	AT1G67580	CDKG2	Cre17.g742250	CDK activating kinase
**CDT1A**	AT2G31270	CDT1	Cre03.g163300	DNA replication initiation factor, CDT1-like
**CYCA1;1**	AT1G44110	CYCA1	Cre03.g207900	A-type cyclin
**CYCB1;1**	AT4G37490	CYCB1	Cre08.g370401	B-type cyclin
**CYCD1;1**	AT1G70210	CYCD1	Cre11.g467772	D-type cyclin
**CYCD2;1**	AT2G22490	CYCD2	Cre06.g289750	D-type cyclin
**CYCD3;1**	AT4G34160	CYCD4	Cre09.g414416	D-type cyclin
**CYCD4;1**	AT5G65420	CYCD5	Cre04.g221301	D-type cyclin
**CYCD6;1**	AT4G03270	CYCD3	Cre06.g284350	D-type cyclin
**CYCT1;2**	AT4G19560	CYCT1	Cre14.g613900	T-type cyclin
**DPA**	AT5G02470	ROC59	Cre10.g425050	Rhythm Of Chloroplast protein 59
**E2F1**	AT5G22220	E2F1	Cre01.g052300	Transcription factor, E2F and DP-related
**E2F3**	AT2G36010	E2F2	Cre13.g572950	Transcription factor, E2F and DP-related
**MAPKKK13**	AT1G07150	MAPKKK13	Cre16.g649100	Mitogen-Activated Protein Kinase Kinase Kinase
**MAPKKK14**	AT2G30040	MAPKKK14	Cre01.g001200	Mitogen-Activated Protein Kinase Kinase Kinase
**MAPKKK5**	AT5G66850	MAPKKK5	Cre07.g339900	Mitogen-Activated Protein Kinase Kinase Kinase
**MAPKKK6**	AT3G07980	MAPKKK6	Cre03.g154250	Mitogen-Activated Protein Kinase Kinase Kinase
**MAPKKK7**	AT3G13530	MAPKKK7	Cre12.g517000	Mitogen-Activated Protein Kinase Kinase Kinase
**MAPKKK9**	AT4G08480	MAPKKK9	Cre12.g545950	Mitogen-Activated Protein Kinase Kinase Kinase
**MCM2**	AT1G44900	n.d.2	Cre10.g436600	K06949–ribosome biogenesis GTPase [EC:3.6.1.-] (rsgA, engC)
**MKK6**	AT5G56580	n.d.	Cre02.g095099	KOG0984–Mitogen-activated protein kinase (MAPK) kinase MKK3/MKK6
**MPK4**	AT4G01370	MAPK8	Cre01.g010000	Mitogen-Activated Protein Kinase
**ORC3**	AT5G16690	n.d.	Cre17.g744247-PF07034	(ORC) subunit 3 N-terminus (ORC3_N)
**PCNA1**	AT1G07370	PCN1	Cre12.g515850	Proliferating cell nuclear antigen homolog
**SMR11**	AT2G28330	PPR4	Cre12.g511400	PPR-Cyclin protein
**SMR4**	AT5G02220	PPR6	Cre10.g437150	PPR-Cyclin protein

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
