# Peer review of "Cross Talk between Hydrogen Peroxide and Nitric Oxide in the Unicellular Green Algae Cell Cycle: How Does It Work?"

_cells, 2022, doi:10.3390/cells11152425_

Round 1
Reviewer 1 Report
This is a review article aimed to discuss the recent updates on potential roles of H2O2 and NO on cell cycle regulation in green algae. I think redox homeostasis is important for the cell cycle regulation. However, many of the suggestions and concerns were not addressed in the revised manuscript. The responses to the raised questions are inadequate. I have a very difficult time searching the edited descriptions for answers to the raised questions.
For response to comment 1, how does modification of the “final conclusion” improve clarity? I guess authors referred to descriptions (in red) added in “8 Summary” on page 14. If that is the case, I don’t think the added description helps. In fact, the added sentences are quite confusing. For example, having genes diurnally regulated during the light/dark cycle does not warrant their functions in cell cycle regulation.
For responses to comment 2 and comment 3, I don’t know where modifications were added and how these modifications answer the raised questions. Taking “Redox dependent components of cell cycle machinery” section as an example, much of the description is still devoted to explain functions of the cell cycle components. It reads like introduction of the cell cycle regulators in Arabidopsis rather than their link to redox regulation. The studies connecting glutathione and the cell cycle in Arabidopsis and Chlamydomonas systems have been published and the references were listed previously. Yet, I don’t see the suggested studies being incorporated in the revised manuscript. As for identification of potential redox-sensitive cysteine, I don’t see the effort of identifying the potential cysteine residues of the Chlamydomonas cell cycle components. You don’t identify the Chlamydomonas protein homologs and just simply assume they have the conserved “cysteine” residues. As for section “ROS and RNS interactions with photosynthesis”, again it reads like chloroplast-mediated redox regulation in Arabidopsis and Chlamydomonas systems. So how are these studies related to the cell cycle control? It is just not clear to me.
Authors did not address my concerns in comment 4. First of all, annotations of Chlamydomonas cell cycle genes were not updated (Table 1, column 4). Secondly, the information on whether Chlamydomonas cell cycle proteins have the conserved and potentially active cysteine residues are still missing.
I still don’t understand why H2O2/NO ratio can be used as an indicator for commercial applicability (in Abstract on page 1) and biomass production (in Summary on page 14). Also, how does the H2O2/NO ratio connect to photosynthesis (page 14)? This issue was raised before but still is not explained explicitly.
What is “commitment competition” (marked in red on page 1 and page 4)?
Many suggested references are still missing in the revised manuscript. For example, references for “Start” and “Restriction Point” (the second paragraph one page 5) are still missing.
On page 7, there is one paragraph dedicated to describe RB/E2F. Since this review deals with the algal systems, I don't understand why authors did not cite the suggested Chlamydomonas references? Also, the Chlamydomonas DREAM complex has been proposed by Chris Benning’s group. Authors also failed to cite their work.
I noticed that the references are not inserted by a reference manager and seem to be typed manually. The obvious mistake is that author’s name is incorrect in the main text but is correct in the reference list. Also, species are not italicized, accents on authors’ names are missing in some but labeled in others.
Author Response
Reviewer 1
Thank you for the comments and suggestions.
This is a review article aimed to discuss the recent updates on potential roles of H2O2 and NO on cell cycle regulation in green algae. I think redox homeostasis is important for the cell cycle regulation. However, many of the suggestions and concerns were not addressed in the revised manuscript. The responses to the raised questions are inadequate. I have a very difficult time searching the edited descriptions for answers to the raised questions.
For response to comment 1, how does modification of the “final conclusion” improve clarity? I guess authors referred to descriptions (in red) added in “8 Summary” on page 14. If that is the case, I don’t think the added description helps. In fact, the added sentences are quite confusing. For example, having genes diurnally regulated during the light/dark cycle does not warrant their functions in cell cycle regulation.
We are very sorry, that the modified content of the summary still does not address Reviewer’s 1 expectations. Since the current, revised text of the summary was accepted by Reviewer 2, and there were no comments addressed to this part of the text from Reviewer 3, we decided not to change it.
For responses to comment 2 and comment 3, I don’t know where modifications were added and how these modifications answer the raised questions. Taking “Redox dependent components of cell cycle machinery” section as an example, much of the description is still devoted to explain functions of the cell cycle components. It reads like introduction of the cell cycle regulators in Arabidopsis rather than their link to redox regulation. The studies connecting glutathione and the cell cycle in Arabidopsis and Chlamydomonas systems have been published and the references were listed previously. Yet, I don’t see the suggested studies being incorporated in the revised manuscript. As for identification of potential redox-sensitive cysteine, I don’t see the effort of identifying the potential cysteine residues of the Chlamydomonas cell cycle components. You don’t identify the Chlamydomonas protein homologs and just simply assume they have the conserved “cysteine” residues. As for section “ROS and RNS interactions with photosynthesis”, again it reads like chloroplast-mediated redox regulation in Arabidopsis and Chlamydomonas systems. So how are these studies related to the cell cycle control? It is just not clear to me.
We are very sorry, that the modifications of the manuscript text still do not address Reviewer’s 1. expectations. Because the improved form of manuscript text was accepted by Reviewer 2, and there were no comments addressed to this part of the text from Reviewer 3, we decides not to change it.
Authors did not address my concerns in comment 4. First of all, annotations of Chlamydomonas cell cycle genes were not updated (Table 1, column 4). Secondly, the information on whether Chlamydomonas cell cycle proteins have the conserved and potentially active cysteine residues are still missing.
As we had mentioned in the previous response to Reviewer’s1 comments, the identification of active cysteine was not the main aim of the data compilation presented in Table 1, and the choice of the database used for annotations of Chlamydomonas genes, including the usage of BLAST originated data, was explained and justified in our previous response. It was not argued by two other Reviewers.
I still don’t understand why H2O2/NO ratio can be used as an indicator for commercial applicability (in Abstract on page 1) and biomass production (in Summary on page 14). Also, how does the H2O2/NO ratio connect to photosynthesis (page 14)? This issue was raised before but still is not explained explicitly.
In the abstract (page 1), the H2O2/NO ratio was changed to the H2O2 or NO level, in the previous round of revision. In summary (page 14), we had left the statement, that the change in the H2O2/NO ratio follows the progress of the cell cycle and its link to photosynthesis is indicated as a possibility, that may require further investigation.
What is “commitment competition” (marked in red on page 1 and page 4)?
The phrase was changed to “commitment competence”, which is more appropriate.
Many suggested references are still missing in the revised manuscript. For example, references for “Start” and “Restriction Point” (the second paragraph one page 5) are still missing.
We are very sorry that this citation was omitted in the previous revision. The appropriate citations were added to the text of the manuscript.
On page 7, there is one paragraph dedicated to describe RB/E2F. Since this review deals with the algal systems, I don't understand why authors did not cite the suggested Chlamydomonas references? Also, the Chlamydomonas DREAM complex has been proposed by Chris Benning’s group. Authors also failed to cite their work.
The text of paragraph was supplemented with information concerning the role of DREAM complex in Chlamydomonas.
I noticed that the references are not inserted by a reference manager and seem to be typed manually. The obvious mistake is that author’s name is incorrect in the main text but is correct in the reference list. Also, species are not italicized, accents on authors’ names are missing in some but labelled in others.
The text of the manuscript, together with the reference list, was checked out for Latin’s names italicization and accents in the names. The required corrections were applied.
Reviewer 2 Report
Reviewer Reports:
I have reviewed the revised version manuscript entitled"Cross talk between hydrogen peroxide and nitric oxide in the unicellular green algae cell cycle: How does it work?"The paper has been improved and can be accepted. I do not have further comments.
.
Author Response
Reviewer 2
Thank you for the comment.
I have reviewed the revised version manuscript entitled"Cross talk between hydrogen peroxide and nitric oxide in the unicellular green algae cell cycle: How does it work?"The paper has been improved and can be accepted. I do not have further comments.
Thank You very much for the positive opinion. We are very glad, that all the manuscript improvements were satisfied for Reviewer 2.
Reviewer 3 Report
The manuscript entitled "Cross talk between hydrogen peroxide and nitric oxide in the unicellular green algae cell cycle: How does it work?" addresses a relevant and appropriate topic for this journal.
Some correction suggestions are given below.
Corrections needed:
line 89/90 - ... and pattern was suggested for Chlamydomonas reinhardtii, ..., and NO peak was recorded for Chlorella (Chlorophyta) species ...
line 93 - algae, such as Heterosigma akashiwo (Ochrophyta, Raphidophyceae), Chaetoceros curvisetus, Skeletonema costatum (Bacillariophyta), Tetraselmis subcordiformis (Chlorophyta), and Gymnodinium sp. (Miozoa) has also been reported ...
line 97 - ... The role of NO was also investigated in the seaweed Gracilaria chilensis (Rhodophyta), ...
line 172/173 - ... Ostreococcus tauri (Chlorophyta) ...
line 461 - ... activity between 24–48 h after ...
Line 550 - Please insert the Figure legend ...
line 561 - ... Within the following 5 h (3 to 8 h of the cell cycle), ...
Author Response
Reviewer 3
Thank you for the comments and suggestions.
The manuscript entitled "Cross talk between hydrogen peroxide and nitric oxide in the unicellular green algae cell cycle: How does it work?" addresses a relevant and appropriate topic for this journal.
Some correction suggestions are given below.
Corrections needed:
line 89/90 - ... and pattern was suggested for Chlamydomonas reinhardtii, ..., and NO peak was recorded for Chlorella (Chlorophyta) species ...
Text was corrected according to Reviewer's suggestion.
line 93 - algae, such as Heterosigma akashiwo (Ochrophyta, Raphidophyceae), Chaetoceros curvisetus, Skeletonema costatum (Bacillariophyta), Tetraselmis subcordiformis (Chlorophyta), and Gymnodinium sp. (Miozoa) has also been reported ...
The text was corrected according to Reviewer's suggestion.
line 97 - ... The role of NO was also investigated in the seaweed Gracilaria chilensis (Rhodophyta), ...
The text was corrected according to Reviewer's suggestion.
line 172/173 - ... Ostreococcus tauri (Chlorophyta) ...
The text was corrected according to Reviewer's suggestion.
line 461 - ... activity between 24–48 h after ...
The text was corrected according to Reviewer's suggestion.
Line 550 - Please insert the Figure legend ...
Figure legend was added as a part of Fig.1 description
line 561 - ... Within the following 5 h (3 to 8 h of the cell cycle), ...
The text of the manuscript was corrected according to Reviewer’s suggestions.
Round 2
Reviewer 1 Report
Even though the authors have taken some of the suggestions and improved the revised manuscript. I regret to say that this manuscript in my opinion remains “Reject”.
Here are the reasons:
1. The authors still failed to add the recommended glutathione-related cell cycle papers (already listed in the first review on 2022/03/22. Below please see the references again.) in the field and did not provide reasons of not citing them.
D. Schnaubelt et al., Low glutathione regulates gene expression and the redox potentials of the nucleus and cytosol in Arabidopsis thaliana. Plant Cell Environ 38, 266-279 (2015).
S. C. Fang, C. L. Chung, C. H. Chen, C. Lopez-Paz, J. G. Umen, Defects in a new class of sulfate/anion transporter link sulfur acclimation responses to intracellular glutathione levels and cell cycle control. Plant Physiol 166, 1852-1868 (2014).
2. The review paper is supposed to provide readers (laymen or experts) a comprehensive and clear overview in the field. Unfortunately, authors declined to make efforts and simply replied as follow.
“ We are very sorry, that the modified content of the summary still does not address Reviewer’s 1 expectations. Since the current, revised text of the summary was accepted by Reviewer 2, and there were no comments addressed to this part of the text from Reviewer 3, we decided not to change it.”
Taking the statement (Page 15 line 623-626) “the expression of nearly 300 genes is under the control of H2O2, where most of them are related to light stress response, and the expression pattern of almost 100 genes follows a day/night oscillation, thus reveals the cell cycle relation. “ as an example, it reads like genes involved in day/night oscillation are related the cell cycle regulation. This is NOT true.
3. Again for Table 1, if authors think the cysteine residues of the cell cycle regulators are important for redox-mediated cell cycle regulation in Chlamydomonas, I still don’t understand why they are reluctant to identify them. The quality of Chlamydomonas gene models are fairly good (thanks to the community efforts). Adding this piece of information would provide information that is unique to green algae. Particularly, the focus of this review is on the unicellular green algal cell cycle.
4. I am still puzzled why authors did not take the advice of using the published annotation for the Chlamydomonas cell cycle regulators (Table 1, column 4). The annotation information on some of the listed components has been published twice – first by Bisová et al (2005) and by Zones et al (2015). I don't understand why authors cited their works but did not follow the nomenclature.
K. Bisová, D. M. Krylov, J. G. Umen, Genome-wide annotation and expression profiling of cell cycle regulatory genes in Chlamydomonas reinhardtii. Plant Physiol 137, 475-491 (2005).
J. M. Zones, I. K. Blaby, S. S. Merchant, J. G. Umen, High-resolution profiling of a synchronized diurnal transcriptome from Chlamydomonas reinhardtii reveals continuous cell and metabolic differentiation. Plant Cell 27, 2743-2769 (2015).
Author Response
We would like to thank the Reviewer for Her/His time and effort to review our manuscript and for all comments and suggestions.
Our responses to the Reviewer’s questions are addressed below one by one.
Even though the authors have taken some of the suggestions and improved the revised manuscript. I regret to say that this manuscript in my opinion remains “Reject”.
Here are the reasons:
- The authors still failed to add the recommended glutathione-related cell cycle papers (already listed in the first review on 2022/03/22. Below please see the references again.) in the field and did not provide reasons of not citing them.
- Schnaubelt et al., Low glutathione regulates gene expression and the redox potentials of the nucleus and cytosol in Arabidopsis thaliana. Plant Cell Environ 38, 266-279 (2015).
- C. Fang, C. L. Chung, C. H. Chen, C. Lopez-Paz, J. G. Umen, Defects in a new class of sulfate/anion transporter link sulfur acclimation responses to intracellular glutathione levels and cell cycle control. Plant Physiol 166, 1852-1868 (2014).
We would like to explain, that the first mentioned work, concerning experiments with A. thaliana (by Schnaubelt et al.), was not previously cited because of the Reviewer's comment that we included too much information about A. thaliana while the text should be focused mainly on algae.
In the present version of the manuscript, both suggested references concerning the role of glutathione pool and its relation to the cell cycle of Arabidopsis and Chlamydomonas were added to the text of the manuscript, according to Reviewer's comment.
- The review paper is supposed to provide readers (laymen or experts) a comprehensive and clear overview in the field. Unfortunately, authors declined to make efforts and simply replied as follow.
“ We are very sorry, that the modified content of the summary still does not address Reviewer’s 1 expectations. Since the current, revised text of the summary was accepted by Reviewer 2, and there were no comments addressed to this part of the text from Reviewer 3, we decided not to change it.”
Taking the statement (Page 15 line 623-626) “the expression of nearly 300 genes is under the control of H2O2, where most of them are related to light stress response, and the expression pattern of almost 100 genes follows a day/night oscillation, thus reveals the cell cycle relation. “ as an example, it reads like genes involved in day/night oscillation are related the cell cycle regulation. This is NOT true.
We are very sorry that in the Reviewer’s opinion we did not make effort to meet Her/His expectations, since we did our best to do that. The revised text of the manuscript was changed significantly (all the changes can be tracked in the changes-tracking mode in the file that was provided to the Editor). While improving our text, we tried to meet the expectations of all three Reviewers thus the prepared version was some kind of “compromise”.
Nonetheless, we do agree with the Reviewer that some parts of the text were not clear enough and could be confusing. Here, we made further changes to the text, to make it more precise. One of the changes concerns the paragraph mentioned in the comment above, about the genes being under the control of H2O2 – we believe that the changed text is more proper.
- Again for Table 1, if authors think the cysteine residues of the cell cycle regulators are important for redox-mediated cell cycle regulation in Chlamydomonas, I still don’t understand why they are reluctant to identify them. The quality of Chlamydomonas gene models are fairly good (thanks to the community efforts). Adding this piece of information would provide information that is unique to green algae. Particularly, the focus of this review is on the unicellular green algal cell cycle.
Indeed, we think that the cysteine residues of the cell cycle regulators are important for redox-mediated cell cycle regulation in Chlamydomonas. That is why the identification of cysteine residues in cell cycle-related proteins is currently under our investigation as a part of a set of genomic and proteomic experiments, indicating the effects of hydrogen peroxide and nitric oxide on the changes in C. reinhardtii cell during its development/cell cycle, with respect to the developmental stage of the cell. The mentioned analyses are planned to be published in our next original paper as an important part of the original, unique data set. Thus, we did not include the data in a review article, which is rather dedicated to presenting the actual overview of the already published results. In the light of the above, we decided to remove the information concerning the no. of cysteine residues from the Table 1 and replace them with genes annotation information, that the Reviewer asked for in point 4.
To be consequent, we also modified the text of the manuscript by removing some parts concerning the role of cysteine residues in cell cycle regulation.
- I am still puzzled why authors did not take the advice of using the published annotation for the Chlamydomonas cell cycle regulators (Table 1, column 4). The annotation information on some of the listed components has been published twice – first by Bisová et al (2005) and by Zones et al (2015). I don't understand why authors cited their works but did not follow the nomenclature.
- Bisová, D. M. Krylov, J. G. Umen, Genome-wide annotation and expression profiling of cell cycle regulatory genes in Chlamydomonas reinhardtii. Plant Physiol 137, 475-491 (2005).
- M. Zones, I. K. Blaby, S. S. Merchant, J. G. Umen, High-resolution profiling of a synchronized diurnal transcriptome from Chlamydomonas reinhardtii reveals continuous cell and metabolic differentiation. Plant Cell 27, 2743-2769 (2015).
Table 1 has been modified. The annotations of genes, that follows the nomenclature used in the papers of Bisová et al. and Zones et al. was added instead of the number of cysteine residues. The headnotes were added to the table, to make it more clear for the reader.
This manuscript is a resubmission of an earlier submission. The following is a list of the peer review reports and author responses from that submission.
Round 1
Reviewer 1 Report
This is a review article aimed to discuss the recent updates on potential roles of reactive oxygen and nitrogen species on cell cycle regulation in green algae. The authors summarized their previous Chlamydomonas studies and attempted to provide synthetics view on regulatory roles of H2O2 and NO on cell cycle regulation in microalgae.
Major concerns:
- It is very difficult to evaluate the important conclusions and/or arguments that the authors want to convey. This review article lacks clear and flowing organization.
- The authors described many studies from Arabidopsis. It is difficult to connect these described works in plants to the current understanding or lack of understanding in microalgae. Besides being photosynthetic, I don’t see any comparison and discussion of the similarity and/or difference of the two evolutionarily disparate systems.
- It is not appropriate to overstate the experimental data. For examples,
- Page 1. “The important role of hydrogen peroxide and nitric oxide in algal cell development as the signals directly involved in the regulation of expression of proteins required for completing of cell cycle like cyclins and cyclin dependent kinases, as well as histone proteins or E2F complex proteins was indicated.” This sentence reads like it is a known fact. But the published studies do not provide adequate evidence to support this statement in the microalgal systems. It is important not to stretch the conclusion that fails to faithfully reflect the nature and range of the current findings. It is however ok to use the data to infer potential regulatory mechanism(s) that require(s) further investigation.
- Page 16, “It was shown that in synchronously growing cells of Chlamydomonas reinhardtii there is a time coincidence between the changes of the H2O2/NO ratio and the occurrence of checkpoints in the cell cycle progression, like G1/S and S/M passage (Pokora et al. 2017). These changes were found to be a result of circadian, cell cycle-related oscillation in the expression of enzymes responsible in Chlamydomonas cells for H2O2 and NO production and scavenging (SOD, CAT, APX, NR).” This is another example of overstatement. There was no genetic data to provide a causal link in “Pokora et al., 2017” study.
- Table 1 does not provide more information than what has already been known. If the exposed cysteines are important, identification of active cysteines of the Chlamydomonas cell cycle proteins would be more informative than just a list of Arabidopsis proteins and their cysteine residues predicted by others. Also, I don’t know how Chlamydomonas orthologs were identified? What criteria were used for BLAST? What is the cut-off value for annotation? Moreover, their analysis includes data that reads like a novel result, but which has been previously published. Because the Chlamydomonas cell cycle genes have already been annotated and published, it is very confusing why the authors came up with another annotation system and did not provide proper attribution for the discoveries and characterizations by other laboratories. This does not conform to the expected scientific community standards.
- Many important studies in the field were not cited in this review article. For examples, annotation of the Chlamydomonas cell cycle genes have been published (1-4), but was not applied in this review. The recent studies reporting redox aspect of the cell cycle regulation in photosynthetic eukaryotes (5-7) were not included.
- Many described studies in the manuscript were not cited. I only list some of them (This is not a full list).
- Page 7. No references on “Start” and “Restriction Point”.
- Page 4. “NO also takes part in environmental stress responses as well as developmental processes in algae” and “The existence of soluble guanylate cyclases (sGC) activated by NO has been documented in Chlamydomonas reinhardtii”.
- Page 6. “Chlamydomonas has metabolic and molecular pathways that are analogous to those in the model plant Arabidopsis (Arabidopsis thaliana) and those that are distinct, offering a glimpse into the diversity of photosynthetic organisms.”
- Page 9. “RBR (retinoblastoma related protein, mammalian RB relative) in plants promote phosphorylation induced by mitogenic signals, through the action of CDKs in association with D-type cyclins, particularly CYCLIN D3:1 (CYCD3:1).”
- The section “Unicellular green algae as model organisms in green cell physiology and toxicology” is not relevant to the topic and can be omitted.
- There is too much irrelevant information in the section of “Redox dependent components of cell cycle machinery”. The relevant perspectives and discussion are missing in this section. This section requires considerable restructure for clarity.
- The section of “ROS and RNS mediated control of cell cycle - evidences in higher plants and algae” also requires re-organization for conciseness.
- Even though the cell cycle is gated to circadian rhythm in photosynthetic eukaryotes, they are defined differently and cannot be used interchangeably.
- Scientific citation requires citation of original work instead of review articles unless conclusions of the review are to be acknowledged. I found many citations are not in line with this standard. Here is one example. Page 9, Tognetti et al., 2017 and Burhans et al., 2009.
- Why did the authors use H2O2/NO ratio as a measurable property to describe redox state?
Other comments:
- The definition of many important concept was not clearly defined. For example, what is the definition of RNS? There is a knowledge gap between RNS and nitric oxide.
- I don’t see references “Pokora et al, 2019” (cited on page 2, 5, 16, and 17) and “Pokora et al, 2015” (cited on page 16) in the reference list.
- The last line on page 5, “allow the cell to attain the commitment competition.” The word “competition” needs to be removed.
- Page 16, what is “HM” treatment? Please define acronyms at the first mention and apply to the whole manuscript.
- The annotation and description of Figure 1 are insufficient.
- Some references were listed twice.
References:
- S. C. Fang, C. de los Reyes, J. G. Umen, Cell size checkpoint control by the retinoblastoma tumor suppressor pathway. PLoS Genet 2, e167 (2006).
- K. Bisová, D. M. Krylov, J. G. Umen, Genome-wide annotation and expression profiling of cell cycle regulatory genes in Chlamydomonas reinhardtii. Plant Physiol 137, 475-491 (2005).
- J. M. Zones, I. K. Blaby, S. S. Merchant, J. G. Umen, High-resolution profiling of a synchronized diurnal transcriptome from Chlamydomonas reinhardtii reveals continuous cell and metabolic differentiation. Plant Cell 27, 2743-2769 (2015).
- Y. Li, D. Liu, C. Lopez-Paz, B. J. Olson, J. G. Umen, A new class of cyclin dependent kinase in Chlamydomonas is required for coupling cell size to cell division. eLife 5, e10767 (2016).
- S. C. Fang, C. L. Chung, C. H. Chen, C. Lopez-Paz, J. G. Umen, Defects in a new class of sulfate/anion transporter link sulfur acclimation responses to intracellular glutathione levels and cell cycle control. Plant Physiol 166, 1852-1868 (2014).
- S. Y. Miyagishima et al., Translation-independent circadian control of the cell cycle in a unicellular photosynthetic eukaryote. Nature communications 5, 3807 (2014).
- D. Schnaubelt et al., Low glutathione regulates gene expression and the redox potentials of the nucleus and cytosol in Arabidopsis thaliana. Plant Cell Environ 38, 266-279 (2015).
Reviewer 2 Report
Reviewer 1:
I recommend major amendments at this level.
General comments:
I reviewed the manuscript entitled “Reactive oxygen and nitrogen species in the regulation of unicellular green algae cell cycle. How can it work?”. The work carried out in the manuscript is interesting which is based on some ROS and RNS, like hydrogen peroxide or nitric oxide, acting as signalling molecules has been demonstrated in relation to some higher plants and algae. Please explain the problem that you want to solve and the contributions of the study in the abstract. The main problem statement and justification for the research have not been clearly stated. Please remove all the multiple references. After that please check the manuscript thoroughly and eliminate all the lumps in the manuscript. This should be done by characterizing each reference individually. This can be done by mentioning 1 or 2 phrases per reference to show how it is different from the others and why it deserves mentioning. This comment is applied all over the paper. It is not clear the contribution of the manuscript to the empirical literature. Would you explicitly specify the novelty of your work? What progress against the most recent state-of-the-art similar studies was made? Discussion of the results should provide useful insights. Please provide research highlights to convey the core findings and provide readers with a quick textual overview of the article. Please, consider numbering the lines next time in order to provide better conditions for reviewers to address the comments. The work looks like only a report and the authors only collect several references without any proper discussion. I am in doubt suitability of the work for this journal at this level. It is recommended that the authors work with a science editor who is proficient in the Native English language to improve the organization and delivery of some portions of the manuscript. This will help improve the readability and help articulate better the relevance of the authors' work. Too many abbreviations are used in the analysis and results. I recommend a nomenclature section for the abbreviations and variables used throughout the passage.
The title doesn't reflect the core of the study. The current title does not interpret the research objectives. Please modify it.
Please improved the abstract. The abstract should have one sentence per each: context and background, motivation, hypothesis, methods, results, conclusions. In the abstract, please add an indication of the achievements from your study that are relevant to the journal scope. Please be concise, maximum 1-2 lines. Data should be incorporated into the abstract. Please explain the problem that you want to solve and the contributions of the study in the abstract.
It is confusing me the title of the introduction?? The authors have only one introduction in the whole manuscript ??? How about another section? Please separate all sections and give proper heading with a number. The literature review is well presented, however, not strongly linked to the gaps of the research, therefore the novelty of the work is not significant. Please improve the state of the art overview, to clearly show the progress beyond the state of the art. The lack of proper justification creates the wrong impression that the authors are unaware of the recent developments. A high-quality paper has to provide a proper state-of-the-art analysis after the literature review and only based on the analysis to formulate the paper goals. The major defect of this study is the debate or argument is not clearly stated in the introduction session. The literature review should clarify the "contribution" of your study. The authors failed to present the study debates and failed to discuss the debates. It is highly recommended to have one table and compare it with others in the same field. In general, the authors should present the specific debate for your study. You may see these articles and follow them in your revised version:
https://www.sciencedirect.com/science/article/pii/S1876610215009583
https://www.sciencedirect.com/science/article/pii/S2214714416306018
https://journals.utm.my/mjce/article/view/15683
Could you please add the aim of the introduction into your revised version?
Please replace the summary with a conclusion. In the conclusions, in addition to summarizing the actions taken and results, please strengthen the explanation of their significance. It is recommended to use quantitative reasoning compared with appropriate benchmarks, especially those stemming from previous work. The conclusion is pretty generic and fails to provide any improvement in the existing knowledge base. Please make sure your conclusions section underscores the scientific value-added of your paper, and/or the applicability of your findings/results. Highlight the novelty of your study.
Bibliography style is not always consistent, please double-check the reference section carefully and correct the inconsistency. References style should follow the journal guideline.